# A Narrative Review of the Classification and Use of Diagnostic Ultrasound for Conditions of the Achilles Tendon

**DOI:** 10.3390/diagnostics10110944

**Published:** 2020-11-13

**Authors:** Sheryl Mascarenhas

**Affiliations:** Department of Internal Medicine, Division of Rheumatology, The Ohio State University Wexner Medical Center, 543 Taylor Ave, Columbus, OH 43203, USA; Sheryl.Mascarenhas@osumc.edu

**Keywords:** ultrasound, enthesitis, spondyloarthropathy, tendonopathy, Achilles tendon

## Abstract

Enthesitis is a cardinal feature of spondyloarthropathies. The Achilles insertion on the calcaneus is a commonly evaluated enthesis located at the hindfoot, generally resulting in hindfoot pain and possible tendon enlargement. For decades, diagnosis of enthesitis was based upon patient history of hindfoot or posterior ankle pain and clinical examination revealing tenderness and/or enlargement at the site of the tendon insertion. However, not all hindfoot or posterior ankle symptoms are related to enthesitis. Advanced imaging, including magnetic resonance imaging (MRI) and ultrasound (US), has allowed for more precise evaluation of hindfoot and posterior ankle conditions. Use of US in diagnosis has helped confirm some of these cases but also identified other conditions that may have otherwise been misclassified without use of advanced imaging diagnostics. Conditions that may result in hindfoot and posterior ankle symptoms related to the Achilles tendon include enthesitis (which can include retrocalcaneal bursitis and insertional tendonopathy), midportion tendonopathy, paratenonopathy, superficial calcaneal bursitis, calcaneal ossification (Haglund deformity), and calcific tendonopathy. With regard to classification of these conditions, much of the existing literature uses confusing nomenclature to describe conditions in this region of the body. Some terminology may imply inflammation when in fact there may be none. A more uniform approach to classifying these conditions based off anatomic location, symptoms, clinical findings, and histopathology is needed. There has been much debate regarding appropriate use of tendonitis when there is no true inflammation, calling instead for use of the terms tendinosis or tendonopathy. To date, there has not been clear examination of a similar overuse of the term enthesitis in conditions where there is no underlying inflammation, thus raising the need for more comprehensive taxonomy.

## 1. Introduction

A principal feature of spondyloarthropathies is inflammation of the tendon, ligament, and joint capsule insertions into the bones, termed enthesitis [1]. Enthesitis can involve many parts of the body, including the insertions of the Achilles, plantar fascia, quadriceps tendon at the upper patellar pole, patellar ligament at the lower patellar pole and tibial tubercle, deltoid at the acromion and clavicle, flexor and extensor tendons at the phalanges, and vertebral ligaments at the spine. [2,3,4,5]. More progressive thinking has led to the model of the enthesis being more than an insertion site but being a unique enthesis organ, including the insertion, the fibrocartilage, bursa, fat pad, adjacent trabecular bone networks, and deeper fascia [6]. 

Clinical assessment of enthesitis can be done by applying ~4 kg/cm^2^ of pressure (enough to blanch the tip of the examiner’s fingernail) and assessing for tenderness [5]. There have been several clinical indices developed to assess enthesial disease activity. The first published index was the Mander/Newcastle Enthesitis Index (MEI), which identified 66 enthesial sites for examination [5,7]. However, assessment of some of these deeper located enthesial areas can be time consuming and also challenging with physical exam alone [8]. Subsequent disease activity indices aimed to condense the examination, with most having less than 16 examination sites. Even with these more focused assessments, the Achilles remained a fixture in these indices, including the Maastricht Ankylosing Spondylitis Enthesitis Score (MASES), the Spondyloarthritis Research Consortium of Canada Enthesitis Index (SPARCC), the Glasgow Ultrasound Enthesitis Scoring System (GUESS), and Leeds Enthesitis Index (LEI) [9,10,11,12].

Diagnosis of enthesitis on clinical exam alone, however, may have limitations as the differential for hindfoot and posterior ankle pain is not exclusive to enthesitis [13,14]. Table 1 includes conditions that may result in pain and/or swelling in the hindfoot and posterior ankle. Some of these conditions, such as an Achilles rupture, are stand-alone conditions not considered a feature of a spondyloarthropathy. Others, such as retrocalcaneal bursitis, may be seen in the context of enthesitis when considering the enthesis as an organ encompassing the bursa and fibrocartilage, for example. Discerning a more accurate etiology to hindfoot symptoms is especially important in diagnosing a spondyloarthropathy. Increasing use of imaging, such as ultrasound (US) and magnetic resonance imaging (MRI), bone scan, and computerized tomography (CT) scan have demonstrated accuracy in detection of both inflammatory and chronic changes in enthesitis [15,16,17,18,19,20]. Identifying the cause of hindfoot pain has important implications in treatment, as different conditions may respond better to medication, exercise, or surgery [21]. Figure 1 illustrates the anatomy and localizes the site of involvement for several conditions that may result in symptoms.

## 2. Methods

In this narrative literature review, databases of PubMed and the Cochrane Library were searched to review the relevant literature. In addition Google Scholar, conference proceedings, and bibliographies of review articles were searched for relevant articles. Key index words were ultrasound, enthesitis, spondyloarthropathy, Achilles, hindfoot, ankle, tendonopathy, tendonitis, paratenon, calcific tendonopathy, retrocalcaneal bursitis, and superficial calcaneal bursitis. Published data from 1927 to 2020 are included in this review.

## 3. A Word about Words

The words used to describe the pathology involving the conditions of the Achilles tendon can be confusing and redundant. Over time, rheumatologists, orthopedists, sports medicine physicians, physiatrists, podiatrists, radiologists, and pathologists have utilized classification schemes that have helped contribute to some of this muddled nomenclature. Reviewing and comparing the literature can be difficult as definitions used by one group may not be the same used by another. 

The first area lacking clarity involves the terminology for describing tendons. In general, “tendonopathy”, sometimes spelled “tendinopathy”, is a catch all term describing pain, swelling, and functional changes in and around tendons [22]. It is an umbrella term that essentially describes tendon pain without specifically denoting the specific pathology; it can include tears, inflammatory enthesitis, or chronic degeneration [23]. When one is not able to differentiate whether the tendon pathology is related to inflammation or degeneration, it may be more accurate to use the word tendonopathy. 

The use of the term tendonopathy to describe non-specific Achilles tendon conditions in the hindfoot and posterior ankle is inhibited in clinical practice. Billing codes are an intrinsic part of clinical documentation; however, the widely used coding system for billing in the US, the International Classification of Diseases-10-Clinical Modification (ICD-10-CM), lacks a billable code for tendonopathy. ICD-10-CM was implemented 1 October 2015; it was designed with a combination of letters and numbers to correspond to a specific condition [24]. Perhaps the lack of consensus on describing tendonopathies, including Achilles tendonopathies, may have led to the noticeable absence in the ICD-10-CM coding system.

There are several codes that approximate tendonopathy, but in general these can be overly inclusive, utilizing descriptors not necessarily seen in tendonopathies alone. For example, M67.89 denotes other disorders of the synovium and tendon [25]. This is too specific for a tendonopathy as selection of this code suggests there is synovial involvement, which is not the case with all tendonopathies. 

While some codes are too specific, some are not specific enough. For example, M67.873 and M67.874 describe other specified disorders of the right or left foot, respectively [26,27]. This unfortunately does not limit pathology to the tendon and is too overly general as this could include other such conditions such as injuries, blood clots, rashes, fractures, bone erosions, ossifications, and tears. 

Some codes used in clinical practice are frankly just inaccurate. Many providers may end up selecting the closest thing they can find to the term tendonopathy, which may be the codes for tendonitis. M76.60, M76.1, or M76.2 denote Achilles tendonitis of an unspecified leg, the right leg, or the left leg, respectively [28]. The suffix “-itis” implies inflammation, and so tendinitis it seems would describe inflammation of the actual tendon. Therefore, use of the tendonitis coding would be inaccurate in describing non-inflammatory, degenerative tendon pathologies.

The traditional use of the word tendonitis among sports medicine practitioners involves a tendon injury, often from a repetitive mechanical load, with a subsequent inflammatory response [29,30]. However, studies have not been able to support evidence for inflammation, at least histologically, within an overloaded tendon [31,32]. In rat models by Zamora et al., in an overloaded model of the rat plantaris tendon, there was no evidence of inflammatory cells on histologic review [33]. This concept that there is a lack of histologic inflammation in tendonitis is further supported in clinical practice, as use of non-steroidal anti-inflammatory medications and corticosteroids have yielded limited long-term success in patients with clinical tendonitis [34]. Therefore, in patients without histologic evaluation, tendonopathy may be a more appropriate descriptor in clinical practice.

The suffix “-osis” implies degeneration; therefore, while tendinitis describes inflammation of the actual tendon itself, tendinosis describes degeneration of the tendon [29]. The collagen degeneration of tendinosis generally occurs in response to chronic overuse [35]. The collagen fibers become disorganized and lose their parallel alignment, becoming loosely packed, interspersed with mucoid ground substance [36]. On histologic examination, tendinosis can result in hypercellularity with increased tenocyctes with myofibroblastic differentiation (tendon repair cells) and neovascularization; however, classic inflammatory cells are generally absent [37]. Like the tendonitis term, the term tendinosis may be better confirmed with histologic evaluation. Therefore, in clinical practice, use of the word tendonopathy may be more befitting without histologic review. 

There are other commonly used classifications which overlap with the above nomenclature, which are similarly disagreed upon in the literature. One of the oldest terms still adopted in practice today is the Haglund terminology. Haglund syndrome was first described almost 90 years ago [38,39]. Haglund syndrome is a description for a retrocalcaneal bursitis associated with or without an abnormal protuberance of the posterosuperior border of the calcaneus [39]. A Haglund deformity or exostosis is a description for the enlargement of the calcaneus; it results from tendinosis, overuse, or the wearing of improperly fitted shoes [40]. Haglund disease instead describes osteochondrosis of the accessory navicular bone [41]. Use of the Haglund nomenclature may be confusing given the diversity of what these terms describe. More recently, some experts have recommended, in lieu of Haglund terminology, adoption of a nomenclature system classified off anatomic location, symptoms, clinical findings, and histopathology [41].

Enthesitis also overlaps with the above terminology and is also not immune to this war of the words. Enthesitis deserves the same linguistic dissection as tendonitis and its counterparts. Similar to the term tendonopathy, enthesopathy is an umbrella term; it describes pathologies of the tendon/ligament insertions, or more progressively, the enthesis organ [42]. Like tendonitis, use of the suffix “-itis” should be reserved to describe actual inflammation of the enthesis. Use of the suffix “-osis” describes degeneration, and so the term “enthesosis” could be considered reasonable to describe degeneration at the enthesis. However, a PubMed search and even a Google search for “enthesosis” yields no results and to the knowledge of the author, this is not a term used in clinical settings. It is important to consider implications of the broad adoption of enthesitis in cases that may not be truly inflammatory. Much like tendinosis, the term enthesosis may be more appropriate in clinical practice when there is a lack of clearly active inflammatory changes.

Much like the Haglund deformity, the enthesophyte (mineralized or ossified scars, or bone spurs) is an often used term that at times gets interchanged with enthesitis or may even be considered by some to be a defining feature of enthesitis [43]. This, however, may be a confusing term as enthesophytes have been found to be a common phenomenon in healthy and aging individuals and may not necessarily imply inflammation [44,45]. Arguably, if the term enthesosis were in use, enthesophytes, which could be a degenerative phenomenon, could be a characteristic of this non-inflammatory condition. 

In addition, greater consideration should be given to describing the anatomy of the enthesis. As noted earlier, the enthesis describes the insertion of tendon and ligaments onto bone with a more recent shift introduced by McGonagle et al. almost 20 years ago, reclassifying the enthesis as a unique organ; this organ includes, collectively, the insertion, the fibrocartilage, bursa, fat pad, adjacent trabecular bone networks, and deeper fascia [1]. Using this broader definition to frame enthesopathies, the term enthesitis should refer to an inflammation within any of these substructures. While this is technically correct, it may be more precise to describe the actual substructures demonstrating pathology when describing enthesopathic findings [46]. This may be more in line with the recommendations put forth by Maffulli et al., with regards to using anatomic location, symptoms, clinical findings, and histopathology to describe hindfoot and posterior ankle conditions in lieu of the Haglund terminology [47]. Specifically, the enthesopathy classification may better be described on the basis of anatomic location with a focus on the specific substructures of the enthesis such as the bursa and Achilles insertion. 

Use of the substructure description of enthesopathies would be important in helping unify the nomenclature of the hindfoot and posterior ankle. It also would help better describe patients with inflammatory conditions. For example, a patient with a retrocalcaneal bursitis, Doppler signal at the enthesis, and erosion could be classified as having enthesitis; arguably, so too could a patient with a thickened tendon alone at the insertion since this could also involve inflammation in the enthesis. However, if each substructure location were described, one could more precisely define and classify patients with posterior ankle and hindfoot inflammation.

For the purposes of this article, the word tendonopathy will be used to refer to pathologies of the tendon and paratenonopathy to pathologies of the paratenon. Further subclassifications of tendinosis, tendonitis, paratenonosis, or paratenonitis would further expand the differential. The word enthesitis will be used when discussing the inflammation at the enthesis and, where possible, further discussion of the substructure locations within the enthesis will be discussed.

## 4. Imaging Modalities for the Hindfoot and Posterior Ankle

Radiographs have limited utility in assessing soft tissue conditions; however, in chronic enthesitis, bony changes, including enthesophytes and erosions, may be seen at the attachments of the Achilles [15]. Radiographs may detect these late stage changes. They also can identify other bony abnormalities in the posterior ankle, including calcific tendonitis and posterior superior calcaneal prominences (Haglund deformities), which can be seen in retrocalcaneal bursitis and insertional Achilles tendonopathy [47,48]. 

Magnetic resonance imaging (MRI) is highly sensitive for active enthesitis; it captures the enthesis as well as associated soft tissue involvement and bone marrow edema [49]. Both exercise-induced tendonopathy and spondyloarthropathies can demonstrate retrocalcaneal bursitis, subcutaneous edema and calcaneal bone marrow edema [18,50,51,52,53]. In spondyloarthropathy patients with severe enthesitis, the thickness of the Achilles tendon is significantly increased [52,53]. In addition to inflammation, MRI is also excellent for detecting full-thickness or partial tendon tears [54].

Medical infrared thermography (MIT) is a less often used imaging modality for the evaluation of tendonopathy; it analyzes physiological function related to blood flow and the control of skin temperature [55]. In relation to tendonopathies, changes in blood flow from neovascularization may affect skin temperature at detectable MIT levels [55,56,57]. There have been limited studies to date evaluating use in the Achilles, but researchers have called for more attention to this non-invasive diagnostic tool [56,58,59,60].

Ultrasound (US) is a highly sensitive and commonly used tool for diagnostic assessments of tendons and entheseal sites [61,62,63,64]. Given the superficial position of the Achilles tendon, and ability of high resolution probes to demonstrate tendon fibers and small structures, US is an imaging modality of choice to evaluate the Achilles tendon [65]. US can depict tendon thickening, enthesophytes, and erosions [66]. US can demonstrate tendon swelling and thickening, discontinuity of tendon fibers, focal hypoechoic intratendinous areas, and fluid around the tendon [67]. In 2004, the Outcome Measures in Rheumatology (OMERACT) US Specialist Interest Group defined enthesitis on US as an “abnormally hypoechoic (loss of normal fibrillar architecture) and/or thickened tendon or ligament at its bony attachment (may occasionally contain hyperechoic foci consistent with calcification), seen in two perpendicular planes that may exhibit Doppler signal and/or bony changes, including enthesophytes, erosions, or irregularity” [68]. Table 2 highlights some of the principle findings on US examination for pathologies related to the Achilles tendon.

Compared to MRI, US remains a highly sensitive test for early diagnosis of enthesitis and generally costs less than MRI [18,87]. It is fast, can be done at the bedside, is reproducible, and is generally a preferred method for assessing tendon pathology among rheumatologists [61,62,63,64,65,66,67,88].

One pitfall in diagnostic US can be improper interpretation of artifacts [89,90]. With regard to the posterior ankle and hindfoot, edge artifact and anisotropy are of particular concern. Anistropy occurs when tissues show abnormal echogenicity, most commonly loss of echogenicity, due to an oblique insonating angle [91]. Tendons or ligaments may appear as hypoechoic and thus could be misinterpreted as tendinosis or tears [92]. Edge artifact occurs when a curved surface reflects the US beam away from the transducer, appearing as hypoechoic parallel lines projecting along the edges of the target [92]. Edge artifact could lead to the curved Achilles tendon being misinterpreted as paratenon thickening [92]. 

## 5. Enthesopathy

The normal disease progression for enthesitis in spondyloarthropathies is purported to begin with local, destructive, microscopic, inflammatory lesions that evolve towards fibrous scarring and new bone formation [66]. The findings on US can include decreased echogenicity of the enthesis, increased dimensions of the enthesis, structural lesions (such as enthesophytes), erosions, and increased vascularity seen on Doppler examination [9,10,11,12,66]. Gandjbakhch et al. reviewed PubMed and Embase databases from 1985–2010 for the most common criteria of enthesitis on US examinations; they found these to include thickened entheses, hypoechogenicity, enthesophytes, bony irregularity at the enthesis, erosions, and surrounding bursitis [69]. Sudoł-Szopińska et al. point out, however, that these criteria are not specific for inflammation and they may originate from chronic damage and degeneration of the enthesis [43]. This finding from damage and degeneration is more aligned with the proposed term enthesosis. Ultimately, this supports the contention that broad use of the term enthesitis may be too far reaching and perhaps use of enthesopathy may be more appropriate in this context when it may unclear if there is truly inflammatory activity present.

Previous studies have looked at the sensitivity and specificity of US in diagnosing enthesitis compared to clinical exam; however, definitions of what constitutes inflammatory enthesitis may not be fully consistent among all studies. Including enthesophytes in the enthesitis scoring, Balint et al. found clinical exam compared to US was less sensitive and specific for enthesitis (22.5% and 79.7%, respectively) [11]. De Miguel et al. utilized the MAdrid Sonographic Enthesis Index (MASEI) to evaluate the diagnostic accuracy of US at the enthesis [93]. The MASEI scoring includes enthesis thickness, structure, calcification/bone proliferation, erosion, bursa, and power Doppler signal in the cortical bone profile, tendon, and bursa. The study looked at 113 early spondyloarthritis patients compared to 57 non-inflammatory control individuals and 24 inflammatory control individuals. The ultrasound score was 23.36 ± 11.40 (mean ± SD) in spondyloarthritis patients and 12.26 ± 6.85 and 16.04 ± 9.94 in the non-inflammatory and inflammatory control groups (*p* < 0.001), respectively. In evaluating US to diagnose enthesitis with the MASEI scoring, the investigators found a sensitivity of 53.1%, a specificity of 83.3% [93].

Much of the previously reported literature on US use for enthesitis is based on varying scoring/classification systems, some of whose findings are not necessarily specific for inflammation but could also be found in degenerative enthesopathies and chronic microinjuries [4,57,94,95,96,97,98,99]. Scoring or classification systems that may include degenerative features such as enthesophytes may more accurately be classifying enthesopathies rather than specific enthesitis. 

There are key findings on US, however, that may help better discern inflammatory enthesitis from a non-inflammatory enthesopathy. Several studies have found the presence of blood vessels in the enthesis to be specific for spondyloarthropathies [3,4,11,98,99]. D’Agostino et al. evaluated entheses of 164 patients with a spondyloarthropathy, 34 with mechanical back pain (MBP) and 30 with rheumatoid arthritis (RA); vascularization at the enthesis was found in 81% of spondyloarthropathy patients but in none of the patients with MBP or RA [100]. While this is not histologic confirmation of inflammation, neovascularization within this region may be the closest finding on US to confirming inflammatory changes.

Poulain et al. further assessed the sensitivity and specificity of power Doppler ultrasound (PDUS) for identifying patients fulfilling the Assessment of SpondyloArthritis International Society (ASAS) classification criteria for axial spondyloarthropathy [101]. Those fulfilling the criteria were deemed ASAS+ and those not fulfilling it were ASAS−. Baseline PDUS was performed at eight entheseal sites with PDUS enthesitis defined by the presence of vascularization at the entheseal insertion. Four hundred two patients with inflammatory back pain underwent a PDUS evaluation; PDUS enthesitis was detected in 58 (14.4%) patients, of which 40 (14.2%) were ASAS+ and 18 (17%) were ASAS−. The sensitivity of PDUS enthesitis was 13.9% and the specificity was 83.5%; the positive predictive value was 69% and negative predictive value was 26.8% for meeting ASAS criteria for axial spondyloarthropathy. Additionally, they found that, of the 18 ASAS− patients with positive PDUS, 59% fulfilled Amor’s criteria, 88% fulfilled European Spondyloarthropathy Study Group criteria, and 59% both [101].

An increasing number of studies have further demonstrated the presence of blood vessels at the tendon insertion in the cortical bone in spondyloarthropathy patients [100,102,103,104]. However, there is no clear consensus on which substructures in these studies demonstrated Doppler signal (e.g., tendon, bursa). This may make it challenging to fully compare studies as what one author may call enthesitis, another may classify the same finding as tendonitis. 

D’Agostino et al. developed a criteria for evaluating what the authors termed enthesitis. The scoring system included points for tendon thickness, hypoechogenicity, calcification, erosion, and vascularization [100]. More specific to enthesitis, the authors semiquantified vascularization with Doppler signal on a scale of 0–3. The vascularization was scored as 0 if Doppler signal was absent, 1 if Doppler signal was minimal (one color spot detected), 2 if Doppler signal was moderate (two spots), or 3 if Doppler signal was severe (≥three spots) [100].

In addition to vascularization, Achilles tendon thickness may also be more specific for enthesitis and may correlate with clinical disease activity indices. In 2017, Ahmed et al. compared US with the Psoriatic Arthritis Disease Activity Score (PASDAS) [105]. PASDAS is a disease activity index for psoriatic arthritis based on patient (PtGA) and physician (PhGA) global, visual analog scale (VAS) scores, tender (SJC66) and swollen (SJC68) joint counts, dactylitis, enthesitis, the physical component score of the short form 36 health survey (SF36-PCS), and C-reactive protein (CRP) level [106]. In comparing 35 psoriatic arthritis patients to 30 matched controls, Achilles tendon thickness in active psoriatic arthritis correlated highly with PASDAS scoring (r = 0.796, *p* < 0.001) [105]. However, it should be noted that the tendon thickness could also be a feature seen in non-inflammatory tendonopathies or enthesopathies.

The Sonographic Enthesitis Index was developed to distinguish between acute and chronic enthesopathy, and identified findings were seen in acute inflammatory enthesial changes and chronic lesions [107]. The authors found increased tendon thickness, hypoechogenicity, peritendinous edema, and bursitis to be more characteristic of acute enthesopathies. Tears, loss of tendon thickness, intratendinous calcifications, and bone erosions were more typical of chronic enthesopathies. With specific regard to the Achilles, the presence of bone erosion was a feature in chronic enthesitis while bursitis was a feature of more acute inflammation [107]. 

US is a useful tool for diagnosing enthesopathy and can help differentiate inflammatory findings of vascularization and tendon thickness that are more often seen in enthesitis. Experts have called for a validated US scoring system for enthesitis not confounded by mechanical factors or obesity [46,108]. Currently used scoring systems may be impacted by body mass index (BMI), for example, MASEI scores, GUESS scores, thickness of the Achilles tendon, and enthesophyte scores correlate with increased BMI [109,110,111,112]. In evaluation of suspected enthesitis, other causes to hindfoot and posterior ankle pain may be elucidated in the workup as outlined in Table 1. The remainder of this article will review some of these other pathologies and how US may help in diagnosis of these conditions.

## 6. Achilles Tendonopathy

Achilles tendonopathy includes tendonitis and tendonosis. On US, the normal tendon morphology is one of parallel hyperechoic striations [70,113]. Tendonopathy may result in alterations in the tendon morphology and/or echogenicity on US [71,72,85]. Echogenicity refers to a tissue’s ability to reflect or transmit sound waves [73]. Hyperechoic structures appear white on the screen, hypoechoic structures appear gray on the screen, and anechoic structures appear black on the screen [72]. In tendonopathies, echogenicity may be decreased from intrasubstance tearing and mucoid degeneration while tendon dimensions may be increased from secondary hypertrophy [74]. Neovascularization may also be seen, as evidenced on US by color power Doppler [75].

Tendonopathy can occur at the insertion of the Achilles tendon or more proximally, at the midportion of the tendon, with the latter being more than twice as common [10,76,77]. Discerning the location of tendonopathy has implications in potential treatment responses. Over 80% of patients with midportion tendonopathy will respond to exercise-based management, whereas only up to 25% of those with insertional Achilles tendonopathy will, and ultimately 47% may go on to need surgical intervention [78,79,80,81]. 

Khan et al. evaluated US accuracy in patients diagnosed clinically with tendonopathy. They demonstrated US had a sensitivity of 0.80 and specificity of 0.49 in diagnosing tendonopathy compared with the clinical exam, and neither color nor power Doppler improved the accuracy of US [82]. Studies have also looked at reproducibility of posterior ankle measurements, particularly the Achilles measurements, and have demonstrated intra- and inter-rater reliability of ultrasound measurement of the Achilles tendon size [114]. Using US to diagnose and evaluate tendon morphology has some limitations, however. One area of uncertainty is what exactly constitutes a normal Achilles measurement, as size variations have been demonstrated even in healthy subjects [115].

There are several risk factors associated with development of tendonopathies. Diabetes and dyslipidemia have been proposed as independent risk factors [116,117]. Higher body mass index (BMI) and older age are associated with higher risk of developing insertional Achilles tendonopathy, with bone deformity, intratendinous calcifications, and distal tendinosis occurring more frequently in individuals with a higher BMI and older age [21,118,119]. Several genetic variants have been proposed, specifically in relation to Achilles tendon injuries, including COL5A1, tenascin C, and matrix metalloproteinase 3 (MMP3) gene [120,121,122].

## 7. Calcific Tendonopathy

For the purposes of this paper, calcific tendonopathies are discussed as a unique entity; however, calcifications are considered a feature of tendonopathies and not a stand-alone condition. The reason for separating out this finding here is to draw attention to the differences in treating calcific tendonopathy compared to non-calcific tendonopathy. Calcifications within the tendon may respond to extracorporeal shock wave therapy, prolotherapy, and surgery [123,124,125]. 

Calcification can occur either at the insertion of the Achilles, termed calcific insertional tendonopathy, or more proximally along the midportion of the Achilles. Dystrophic calcification or ossification may appear on US, appearing as echogenic collections with posterior acoustic shadowing [74]. Calcific insertional tendonopathy is characterized by ossification of the enthesial fibrocartilage with ossification or bone spur formation at the insertion of the tendon [40]. Figure 2B illustrates an example of US findings of calcific tendonopathy.

Given that calcifications of the tendons are not usually classified as a separate entity in the US literature, data on actual specificity and sensitivity of US in assessing calcific tendonopathies as a stand-alone finding is limited, especially with regard to the Achilles tendon. However, there are studies demonstrating high sensitivity for US in identifying shoulder calcific tendonitis [126,127].

## 8. Retrocalcaneal and Superficial Calcaneal Bursitis

Bursa are synovium-lined, sac-like structures located near bony prominences and between bones, muscles, tendons, and ligaments; bursitis refers to swelling or inflammation of the bursa sacs [128]. Bursitis may occur from overuse, infection, trauma, and inflammatory disorders [129]. There are two noteworthy bursa in the hindfoot, the retrocalcaneal bursa and the superficial calcaneal bursa. Figure 1 illustrates that the retrocalcaneal bursa is deep to the Achilles tendon, adjacent to the calcaneus and the superficial calcaneal bursa is superior to the Achilles tendon. Figure 2D is an example of US findings of retrocalcaneal bursitis.

The retrocalcaneal bursa is a constant bursa that can be visible on US in 25% of healthy individuals [130]. In 1998, Olivieri et al. evaluated the diagnostic accuracy of US in demonstrating hindfoot bursitis [53]. They evaluated 14 patients meeting Amor criteria for spondyloarthropathies. Using MRI as the gold standard, they found that while US had a high specificity, it surprisingly had low sensitivity. They reported US had 50% sensitivity and 100% specificity [53]. 

## 9. Paratenonopathy

Unlike most other tendons in the body, the Achilles is surrounded by a paratenon [10,131,132,133,134,135]. A paratenon is loose, vascular, areolar connective tissue surrounding the tendon with a thin layer of synovial cells [37,135,136]. The paratenon is adjacent to the Achilles tendon, located approximately 4–6 cm above the calcaneus; it helps supply blood to the Achilles tendon [136,137,138]. Paratenon abnormalities are more frequently observed on the medial aspect of the Achilles tendon [138]. On US, the paratenon may appear as hypoechoic edematous fat and fluid extending several centimeters craniocaudally [75]. 

There are limited investigations reviewing ultrasound diagnostic accuracy of paratenonopathies. In 2015, Stecco et al. compared paratenon thickness between healthy and symptomatic individuals [139]. They found the mean value of the paratenon in a normal subject of 0.95 mm (SD 0.16) compared to symptomatic patients who had a mean value of 1.27 mm (SD 0.29). The ultrasound evaluation demonstrated a statistically significant difference of paratenon thickness between normal subjects and symptomatic patients (*p* = 0.0005). The authors also still found a significant difference in paratenon thickness after controlling for BMI (*p* = 0.041) [139]. 

## 10. Achilles Tendon Tears and Muscle Ruptures

On US, intratendon tears appear as discretely margined defects within the tendon itself [85]. Fluid may be visualized separating the torn margins of the tendon fibers; these may exhibit edge artifact, also known as diffraction shadowing [75,86]. Figure 2C is an example of an intrasubstance Achilles tear. Plantaris ruptures can mimic the appearance of paratenonitis on ultrasound with hypoechoic edematous fat and fluid extending craniocaudally; however, the torn tendon ends can be seen at the margins of the edema [75]. 

When compared with surgery, US is sensitive and specific for detecting tendon tears [140]. It can also further differentiate partial- from full-thickness tears in the Achilles tendon [133]. Ultrasound has some limitations, however, in differentiating a partial Achilles tendon rupture from a discrete area of tendinosis [141]. Data on diagnostic accuracy of US in diagnosing Achilles ruptures ranges from 79.6% to 100% [60,125,142,143,144]. 

Griffin et al. determined that performing the Thompson test during the real time Achilles ultrasound test (RAUT) is more sensitive and more specific than traditional (static) ultrasound [142]. For static ultrasound, accuracy was high with a sensitivity and specificity of 76.8% and 74.8% for the novice reviewers and 79.6% and 86.4% for attending reviewers, respectively. The incorporation of RAUT testing increased this further, with a sensitivity and specificity of 87.2% and 81.1% for the novice group and 86.4% and 91.7% for the attending group, respectively [142].

Clinically, Achilles tendon ruptures are more commonly seen in people participating in running, jumping, and agility activities involving eccentric loading and explosive plyometric contractions [145]. In runners, increased femoral anteversion, leg length discrepancy, muscle weakness, increased body mass index, and older age increase intrinsic risk of Achilles tendon rupture [145]. Extrinsic risk factors for tendon tears and ruptures of the Achilles include fluoroquinolones and corticosteroids [146]. 

## 11. Conclusions

Ultrasound imaging has proven to be an important tool that can help in localizing more precisely the source of symptoms in the hindfoot and posterior ankle. Pain and/or swelling in this region has a broad differential, focused largely on the Achilles tendon or its adjacent structures. With US, it is possible to visualize these areas. With regard to the enthesis organ, US can further help identify substructures and determine the extent of their involvement. This may help further characterize and classify individuals with enthesopathies. Use of color Doppler may help further identify features more consistent with inflammatory conditions, including enthesitis. 

A critical issue with the literature has been inconsistency in describing pathologies of this region. Adoption of a uniform nomenclature involving anatomy and histologic references would be an important step in advancing further research and clinical practice.

## Figures and Tables

**Figure 1 diagnostics-10-00944-f001:**
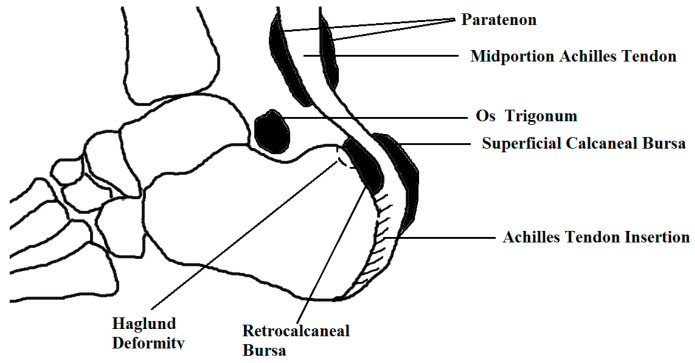
Hindfoot and posterior ankle anatomy.

**Figure 2 diagnostics-10-00944-f002:**
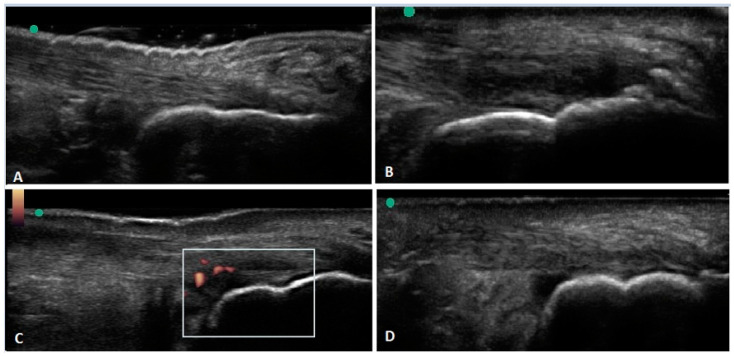
Ultrasound imaging of Achilles tendon. Figure 2 illustrates several examples of ultrasound findings in the posterior ankle and hindfoot. (**A**) Normal imaging of hindfoot/posterior ankle. (**B**) Insertional Achilles calcific tendonopathy. (**C**) Intrasubstance Achilles tear with surrounding neovascularization. (**D**) Retrocalcaneal bursitis. Note the green dot indicates probe orientation. The box in C is the color box indicating the region of color Doppler assessment.

**Table 1 diagnostics-10-00944-t001:** Etiologies for hindfoot and posterior ankle pain and/or swelling.

Differential Diagnoses for Conditions of the Achilles Region
Midportion Achilles tendonopathy
Insertional Achilles tendonopathy *
Achilles paratenonopathy
Midportion Achilles calcific tendonopathy
Insertional Achilles calcific tendonopathy *
Enthesopathy
Retrocalcaneal bursitis *^,¶^
Superficial calcaneal bursitis
Os trigonum syndrome
Tophaceous gout
Calcium pyrophosphate deposition disease
Achilles tendon xanthomata
Ruptured gastrocnemius
Ruptured plantaris
Ruptured popliteal cyst with extravasation down gastrocnemius
Calcaneal ossification (also known as Haglund deformity) ^¶^

Note: the above terms with the suffix -opathy include their respective -itis and -osis subclassifications. * Features of enthesitis. ^¶^ Features of Haglund syndrome.

**Table 2 diagnostics-10-00944-t002:** Ultrasound findings for Achilles tendon pathologies.

Condition	Ultrasound Findings	References
**Enthesopathy**	Decreased echogenicity of the enthesis Increased dimensions of the enthesis Structural lesions (such as enthesophytes) Erosions Increased vascularity seen on color power Doppler	[9,10,11,12,66,69]
**Tendonopathy**	Echogenicity may be decreased Tendon dimensions may be increased from secondary hypertrophy Neovascularization seen on color power Doppler Structural lesions (such as calcifications)	[70,71,72,73,74,75,76,77,78,79,80,81,82]
**Bursitis**	Anechoic structure Hyperechoic lining	[83]
**Paratenonopathy**	Thickened paratenon Neovascularization on color power Doppler Increased echogenicity of pre-Achilles fat pad	[75,84]
**Tendon tear**	Margined defect within the tendon itself Fluid may be visualized separating the torn margins of the tendon fibers, which may exhibit edge artifact	[75,85,86]

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
