# Peer review of "A Narrative Review of the Classification and Use of Diagnostic Ultrasound for Conditions of the Achilles Tendon"

_diagnostics, 2020, doi:10.3390/diagnostics10110944_

Round 1
Reviewer 1 Report
- The major concern is the title. The diagnosis of the hindfoot should include the subtalus joint, ligament, plantar fascia, and sinus tarsi. The title should be modified. The reference can be used as the consideration: Advanced Ankle and Foot Sonoanatomy: Imaging Beyond the Basics. Diagnostics (Basel). 2020 Mar 14;10(3):160.
- In this article, no images of the Ultrasound. I suggested adding some figure acording to the tendinopathy, tendon rupture, retrocalcaneus bursa, Os Trigonum, and spurs at heel.
- In line 55-56, not only the MRI and US can be used at enthesitis, the bone scan, CT can also be used. Please add some sentence and reference.
- The artifact of the ultrasound shall be mentioned, which can lead to wrong diagnosis for the normal Achilles tendon and tendinopathy. The reference : Artifacts in Musculoskeletal Ultrasonography: From Physics to Clinics. Diagnostics (Basel). 2020 Aug 27;10(9):E645.
- The line 366, muscle rupture: soleus or gastrocnemius? the plantaris at this region is only the tendon part, and may involve into the Achilles tendon.
Author Response
- I adjusted the title to focus on Achilles tendon classification and diagnositics
- Figure 2 added for additional imaging of US (line 55-56)
- I have added the references
- I have added a paragraph discussing some of the US artifacts (line 199-206)
- This sentence is originally in a confusing position. It should be in reference to the Achilles tendon tear. I have moved it to the next paragraph where I think it makes more sense. conditions in this region of the body. A more uniform approach to classifying these conditions based of location
Reviewer 2 Report
Thanks for the opportunity to review this manuscript entitle "Diagnosis of Hindfoot and Posterior Ankle Conditions with Ultrasound". I provide you some major concerns that should be accurately addressed in the present work:
-The title needs to reflect the study design as a narrative review.
-The abstract should summarize the main conclusions of this study. The future verb tense is more appropiate for a project.
-After the introduction, a methods section should be included and divided into different subsections like study design (please, see PRISMA criteria), search strategy, inclusion and exclusion criteria, risk of bias... In summary, all subsections recommended by the PRISMA criteria should be accurately addressed. Please, see https://www.equator-network.org/reporting-guidelines/prisma/
-In addition, flow diagram and checklist recommended by these criteria should be provided in the review.
-The results should include tables sumarizing the findings of this study in an accurate way.
-This topic is very interesting, but an important work should be carried out before considering it for publication in order to adapt this review to the scientific presentation recommended by PRISMA criteria.
Author Response
Thank you for your feedback
- I have adjusted the title to reflect the Narrative Review
- I have changed the verb tense to avoid use of future tense and expanded the abstract section further to include the taxonomy as well
- The PRISMA comments are well taken, but would be more applicable to a systematic review or meta-analysis—which this paper is not. As a narrative review this paper highlights several key studies evaluating US use in the hindfoot and ankle, and may likely raise further need for a future systematic review regarding this topic
- I have added Table 2 to include a summary of the US findings
Reviewer 3 Report
Original US imaging are required to show all difference that has been commented on the paper. Image caption are mandatoried to explain all images.
You have to modify the title to show the review type of the present manuscript
summary tables with esencial author consulted in the article showing the different characteristics evaluated for each one are required to leave a clear sight to lectors.
Author Response
Thank you for your feedback
- US images have been added to the paper
- Title has been modified
- Table 2 has been added to include the US findings
Round 2
Reviewer 2 Report
Authors have not adressed my prior requirements. Despite the PRISMA criteria may be more adequate for a systematic review or meta-analysis, the narrative review could have been improved with a systematic bibliographic search.
Author Response
Thank you for your feedback.
- I have added a methods section describing the search methods for the narrative review.
Reviewer 3 Report
It would be appropiate to add some references of termography as another valid tool of all Achilles tendinopathies described on the present work
There is a mistake on initials MASAI, instead of MASEI
I missed a flow chart where I could see the selection criteria of the author to chose the papers selected and a paragraph where the author talked about these screening criteria
Author Response
Thank you for your feedback.
- I have added a paragraph referencing thermography as an imaging modality for the Achilles in section 4.
- I have corrected the acronym to MASEI
- I added a methods section describing the search methods for the narrative review
Round 3
Reviewer 2 Report
Again, authors have not followed my recomendations. Despite this study is a narrative review, the inclusion and citation of PRISMA criteria should be carried out in order to improve the study reproducibility:
-1. Include all subsetions of PRISMA criteria and justify the limitations for the subsection that were not included in the methods section. Cite the PRISMA criteria with an appropiate reference.
-2. The PRISMA checklist should be added as a supplemental file.
-3. The PRISMA flow diagram should be added.
-4. Overall manuscript should be divided into introduction, methods, results, discussion and conclusions.
-5. The abstract should includy search strategy, study type... please, see PRISMA criteria.
I would like to acept your manuscript, but if you do not follow my recommendations, I can not acept it in its current form. Thanks.